

# History of the Tromsø Ionosphere Heating facility

Michael T. Rietveld[1,2], Peter Stubbe[3]

[1] retired from EISCAT Scientific Association, Ramfjordmoen, 9027 Ramfjordbotn, Norway
[2] retired from Institute for Physics and Technology, UiT The Arctic University of Norway, 9037 Tromsø, Norway
[3]retired from Max-Planck-Institut für Aeronomie, Katlenburg-Lindau, Germany

*Correspondence to*: Michael T. Rietveld (mikerietveld20@gmail.com)

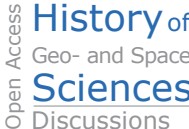

**Abstract.**

We present the historical background to the construction of a major ionospheric heating facility near Tromsø, Norway in the 1970s by the Max Planck Institute for Aeronomy and the subsequent operational history to the present. It was built next to the EISCAT incoherent scatter radar facility and in a region with a multitude of diagnostic instruments used to study the auroral region. The facility was transferred to the EISCAT Scientific Association in January 1993 and continues to provide

new discoveries in plasma physics and ionospheric and atmospheric science to this day. It is expected that 'Heating' will continue operating together with the new generation of incoherent scatter radar, called EISCAT_3D, when it is commissioned in the near future.

**Introduction**

In the following we present the history of a major ionospheric research facility which played a very important part of both of

our scientific careers. The second author was involved right from the start of the project until the transfer of the facility from the Max Planck Institute for Aeronomy to EISCAT in 1993. We worked together for several years up to this date, after which the first author managed the facility until his retirement in 2020. This history, which concentrates on the administrative and technical aspects but also mentions important scientific collaborations and results, is based on our personal memories and documents available to us but which may not be easily accessible to all readers.

**1 Background and Conception**


The history of ionospheric heating experiments started in the early days of radio, with the Luxembourg effect (Tellegen, 1933) where the modulation of a powerful radio transmitter was imparted in the ionosphere on another radio wave transmission. The explanation was that the powerful radio wave could heat the free electrons in the plasma which makes up the ionosphere, and thus change the properties of the medium. The ionosphere is the ionised part of the upper atmosphere,

extending from about 70 km to several hundred kilometres. By injecting high-power radio waves into this plasma it became possible to use the ionosphere as a plasma laboratory without the restriction of boundary walls which limit some laboratory plasma experiments. A better understanding of plasma physics was made possible as well as providing a new technique to learn more about the ionosphere itself. A major advance was made in the early 1970s with a heating facility in Boulder, Colorado, US, which showed a plethora of unexpected ionospheric phenomena which could be triggered by powerful high

frequency (HF) waves (*Radio Science*, 9, 11, 1974). This spawned international interest especially in the Cold War era where radio communication via the ionosphere was still important and ionospheric irregularities played an increasingly important role in understanding radar echoes from natural phenomena as well as man-made objects. At the same time the incoherent scatter radar technique for measurement of ionospheric plasma properties was developing rapidly with US facilities in Arecibo, Puerto Rico, Millstone Hill and Chatanika, Alaska, and European radars in France and UK either operating or being

planned. An advanced tristatic European radar was in the planning for northern Scandinavia to investigate the auroral
ionosphere. The Max-Planck Society, one of the major scientific research organisations in Germany, was already a partner in
the planning of this radar and became one of the six associates in the European Incoherent Scatter Scientific Association
(EISCAT). Two Max Planck institutes were involved here, The Max Planck Institute for Extraterrestrial Physics in Garching
and the Max Planck Institute for Aeronomy (MPAe) in Katlenburg-Lindau, now Max Planck Institute for Solar System

Research, in Göttingen (Haerendel, 2016). Ian Axford, who had just been appointed as a new director of MPAe in 1974,
strongly supported and influenced the development of EISCAT and suggested building an ionospheric heating facility.
About this time similar facilities were being built near Arecibo and in the former Soviet Union, both at low to mid-latitudes.
The MPAe had a long history in the area of HF radio research and techniques for the purpose of studying the ionosphere and
upper atmosphere (e.g., Czechowsky and Rüster, 2007) such that designing and constructing a high-power HF transmitting

facility was well within the competence of the institute. With the new director (Axford) came a number of international guest
researchers who helped develop the science case for an ionospheric heater as well as advised on technical issues. Some of
these were Fred Hibberd from Australia, Jules Fejer from USA (who was also an external scientific member of MPAe) and
Dick Dowden from New Zealand. The project was led by Prof. Peter Stubbe and Dipl. Phys. Helmut Kopka. Other scientists
from MPAe who were closely involved with "Project Heating", as it was called, were Prof. Harry Kohl, Dr. Gerhard Rose,

and Dipl. Phys. Hans Lauche.

**2 Funding, Construction and Inauguration**

Planning and design of "project Heating" started in 1975. The total cost of the facility was estimated to be 6 M German
Marks (DM) at that time, equivalent to 3 M € Approximately a third of this amount (for development, operating and
personnel costs) came from the operating budget of MPAe, one third for investment from the Max Planck Society, and the

remaining third for investment from the German Research Foundation (Deutsche Forschungsgemeinschaft, or DFG). This
amount did not include the salary costs of the staff who worked on the project at MPAe. The project was formally a joint
project between MPAe and the University of Tromsø (UiT) where funding, design and construction was the responsibility of
MPAe, and UiT, especially through Asgeir Brekke and Reidulv Larsen, arranged local permits and infrastructure such as the
buildings. UiT was also the main Norwegian partner in EISCAT (See Holt, 2012). For several reasons the site was chosen to

be next to the new EISCAT radars in Ramfjordmoen, about 30 km by road south of Tromsø. MPAe and UiT were already
involved in the EISCAT radar project which would be a major diagnostic for Heating, and there were many common
infrastructure items such as communication and the electric power line which was specially installed to supply both projects.
An important feature of the location in the arctic as compared to mid- or low-latitudes is that the Earth's magnetic field is
close to vertical which is advantageous for studying many plasma-physical phenomena. Figure 1 shows an aerial view of the

heating facility next to the EISCAT radars. A history of the EISCAT system was recently published by Wannberg (2021).

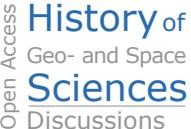

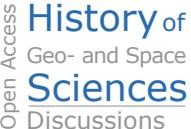

Figure 1: Aerial photo of the Heating antennas and incoherent scatter radars at the Ramfjordmoen site from 1996. (Photo: Fjellanger Widerøe AS)


Design concepts for the facility were outlined by Kopka et al. (1976) and these were largely adhered to in the final facility. The design of the facility was challenging. To cover the wide frequency range from 2.7 to 8 MHz three antenna arrays, each of 6 x 6 crossed full wave dipoles were built to the same design but the lengths, spacing and height above ground were scaled by √2 between arrays. The full-wave antennas were rhombically broadened to provide a wider bandwidth than single

wire antennas. The centre frequency was 3.31 MHz for Array-1, 4.71 MHz for Array-2 and 6.63 MHz for Array-3, each with a 37% bandwidth. Each array was fed by high power coaxial cables, with a total length of ca. 50 km, from a central building housing 12 transmitters of up to 100 MW each. The gain of each array was 24 dBi at the mid-frequency (dBi is the maximum gain of the radiation pattern in dB compared with that of an isotropically radiating antenna) resulting in a maximum effective radiated power (ERP) of 300 MW. Figure 2 shows a schematic diagram of the antenna arrays together

with other relevant instruments and buildings in the Ramfjordmoen site. Commercial high-power coaxial cable and other
components were too expensive so that in-house designed and built air-filled coaxial cables, baluns, power splitters, quarter-
wave transformers, stubs and motor-driven coaxial switches were produced from aluminium pipes and specially-designed
castings and fabrications (see Fig. 3). This was a highly innovative but also risky undertaking. The design worked very well
electrically, but required retro-fitting of an air-dryer and compressor to feed dry air into the whole system to prevent ingress

of moisture, as well as many more wooden supports under the coaxial lines in order to survive the harsh winter conditions of
northern Norway. The accumulation of up to 2 meters of snow and ice during a winter could bend, deform or break some of
the aluminium components, and the subsequent thaw in the spring meant that parts of the lines could be under water. Nearly
all the connectors are in aluminium so the prevention of moisture ingress in the transmission lines is very important, which
the compressor and dryer achieved successfully. In spite of the air dryer and extra wooden supports under the cables,

maintenance of this coaxial cable feed system would remain a fairly labour-intensive annual task each summer.

A conducting ground plane was never installed as it was felt to be unnecessary given the moist ground conditions at
Ramfjordmoen, so the calculated ERP assumed a perfectly conducting ground. In hindsight, a ground plane might have
increased the actual ERP since subsequent modelling reported in Senior et al. (2011) suggests that, using measured values of

ground conductivity and dielectric constant, the actual ERP is approximately 75% of that calculated assuming a perfect
ground.

The transmitters were state of the art vacuum tube transmitters with a wide-band solid state driver and automatic tuning and
impedance matching system employing variable vacuum capacitors and switchable inductors between frequency bands. The

main power amplifier was a linear, water-cooled Siemens tetrode RS2052CJ vacuum tube with variable voltage power
supplies using thyristors. A prototype design from Siemens was used to build the transmitters including power supplies in-
house. The wide-band solid-state driver was designed to deliver ca. 1.5 kW to the power amplifier tube and was built in-
house. It turned out that the impedance matching to the tube was not good so that an impedance-matching filter needed to be
designed and constructed, which turned out to be the equivalent of an engineering masters thesis (Diplomarbeit in German).

Figure 4 shows some of the open transmitter cabinets together with three persons who were closely involved in the early part
of the project.



Figure 2: Schematic diagram showing to scale the layout of antennas and major infrastructure at the Ramfjordmoen site. The red diagonal lines in the HF arrays represent the full wave antennas attached to wooden masts at each end. The feed points of each crossed dipole antenna is where the red lines cross each other. The black squares in HF Array‑1 show the original 23 m tall wooden masts to which the low frequency antennas were attached (until 1985), and the red crosses in Array‑1 show the 12 m tall wooden masts added around 1989 for the modified array containing higher frequency antennas similar to Array‑3. The Dynasonde (HF sounder), housed in the same building as the HF control room, and its associated antennas are an integral part of the heating facility. The EISCAT incoherent scatter radar (ISR) antennas and buildings are shown at the upper right. The large blue crosses show the crossed half-wave dipoles of the 2.78 MHz MF radar transmitting antenna, suspended between masts (small blue crosses). The ionosonde tower supports the transmitting antenna for the University of



Tromsø's Digisonde. The Morro array is an antenna for a 56 MHz MST radar from the University of Tromsø. The green double line shows the road, and the unlabelled boxes are buildings or huts with optical and other instruments.


Figure 3: The fabricated coaxial cable and towers in one row of antennas of Array-2 in Spring of 2010. The vertical tubes which are part of the tower supporting the antenna are also coaxial cables acting as quarter wave transformers supplying the RF current to the antenna. Array-3 has an identical design but is smaller in size. Originally Array-1 had similar antennas, larger in size, but these were modified after the storm in 1985. (Photo M. T. Rietveld)

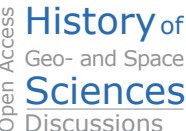

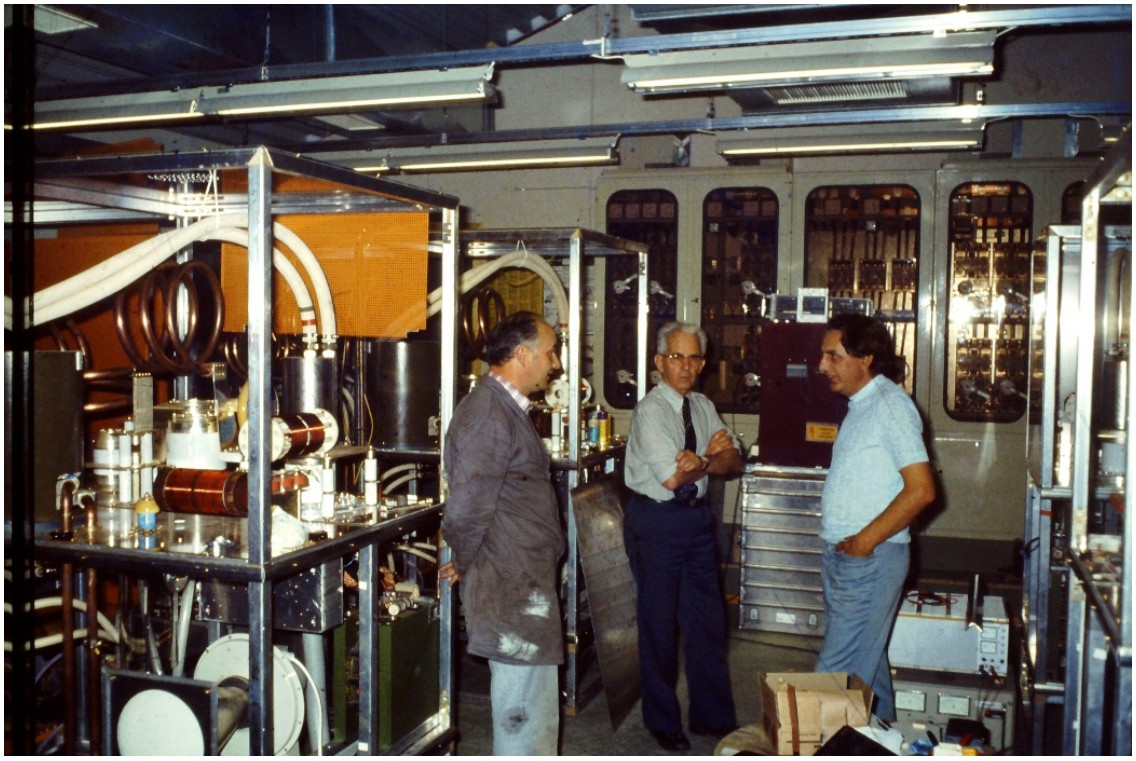

Figure 4: Three important people involved in the construction of Heating. From left technician Helmut Gegner, and scientists Jules Fejer and Helmut Kopka in the transmitter hall in 1979. Four open 100 kW transmitters are visible, which were used in
the first experiments. (photo MPAe)

The RF waveform was produced using the then-new HP3325A synthesizers, with one for each of the 12 transmitters and an additional one as phase reference. These were computer controlled from a Commodore PET microcomputer running a combination of assembler and BASIC language programs with a specially developed interface to the essential transmitter
hardware. For modulation of the transmitter output another microcomputer, a Texas Instruments TM 990 programmed in assembler language, was used with 12 digital to analogue converters and other special hardware to provide external control voltages to the HP synthesizers to vary their amplitude and phase. Low frequency synthesizers could also be used to amplitude modulate the RF waveform from the HP synthesizers. Figure 5 shows a block diagram of the transmission system and the power distribution to the various antenna arrays.




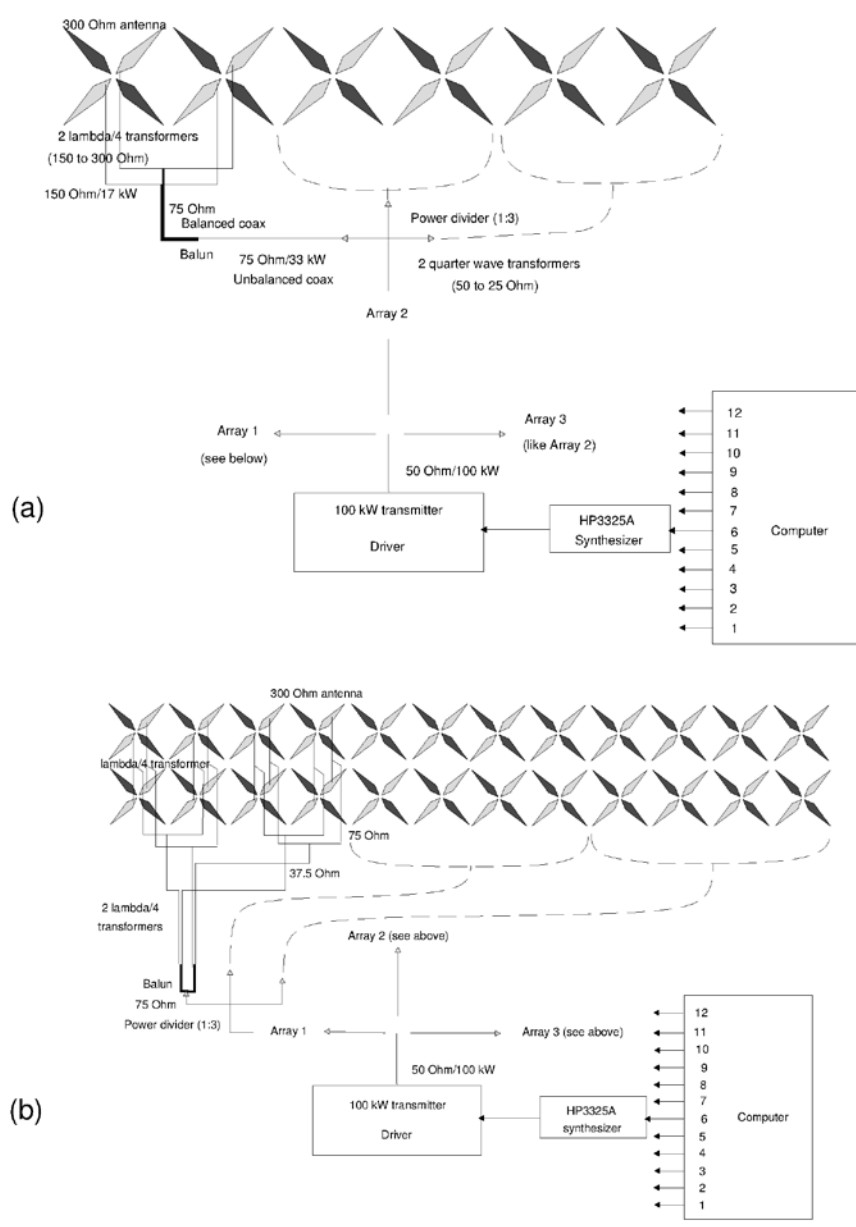

Figure 5: Diagram showing the transmission system and how the power was distributed from one transmitter (a) to six crossed dipoles in each of the antenna arrays before the storm in 1985, and (b) to 24 crossed dipoles in the rebuilt Array-1 from 1990. After 2008 the HP3325A synthesizers were replaced by direct digital synthesizer boards.


Construction of the facility in Ramfjordmoen occurred between 1978 and 1980. This involved long periods of work by teams of up to about 10 engineers, technicians and other workers from MPAe who stayed in the main accommodation and control building in Ramfjordmoen. In anticipation of large teams having to stay for long periods, this building was designed to be relatively spacious and well-equipped. MPAe staff who designed and built parts of the facility were Dr. Rainer Kramm (electronics and software), Richard Zwick and Lothar Bemmann. H.-G. Kellner, E. Schäfer, K. Schreiber, R. Pabst, and W. Butscheck built and assembled much of the hardware and Helmut Gegner and Klaus Eulig assembled, maintained and operated the facility for the duration of the project under MPAe ownership. Many other workers from MPAe were involved in the construction of parts in Katlenburg-Lindau and subsequent assembly work in Tromsø for shorter periods.

In the control building there was also installed an advanced HF radar developed at the National Oceanic and Atmospheric Administration (NOAA, Boulder CO), called the Dynasonde. This instrument was and still is essential to determine the state of the ionosphere continuously and in real time, independent of the EISCAT ISRs which are not always operating. It was also an important diagnostic instrument to measure the effects of heating the ionosphere. J. W. (Bill) Wright, and R. Grubb were valuable collaborators in the set-up and use of this versatile instrument. The computer hardware, operating and analysis software were later upgraded such that this instrument is still providing advanced high-quality ionospheric data to this day (Rietveld et al., 2008).

There was an official inauguration in September 1980, by the director of MPAe, Ian Axford, and the rector of the University of Tromsø, Prof. Yngvar Løchen, and as guests from USA were Bill Gordon and Tor Hagfors, then director of EISCAT, both pioneers in the ionospheric heating field. At this stage there had already been experiments performed using the available, finished transmitters out of the final twelve transmitters. The first experiments were with the Norwegian partial reflection (PRE) system of UiT (August 1979), followed by the first VLF excitation experiments with Dowden's system (March 1980), excitation of micropulsations with Lotz and Watermann (September 1980), and the first anomalous absorption experiments with Tudor Jones' group from the University of Leicester, UK (October 1980). These first experiments are described in summarized form in Stubbe et al. (1980). Another early result from August 1980 was the measurement of fast attenuation of the reflected HF wave which was explained in terms of the parametric decay instability and a slower attenuation by heater-induced striations (Fejer and Kopka, 1981). The original duration of the Heating project was limited to the end of 1987 but, because of the successful results obtained, it was decided to extend the project until the end of 1992.

A comment is in order about the name of the facility. At MPAe it was a project, simply called "Heating" because of the main physical effect of heating the electrons in the ionosphere which it could produce. This was not an acronym although it has sometimes been spelled with all capital letters. After the transfer to EISCAT it was also called Heating or the heating facility.

Because thermal heating of electrons is not the only effect of the powerful HF wave, it is sometimes simply called the HF-facility or HF-pump because the HF waves can excite and pump various plasma instabilities, and can thereby energise electrons to supra-thermal energies.

## 3 The first decade of discoveries

The results of the first 2-3 years were quickly published and summarised in a review paper by Stubbe et al. (1982) followed
by another review three years later (Stubbe et al. 1985). Many important scientific discoveries were made in the first 10 years of operation. One of the highlights was the exploration of the process by which ionospherically-induced currents in the lower ionosphere generate extra-low to very-low frequency (ELF, VLF) radio waves in the audio-frequency range and below.  With measurements from ELF/VLF and micropulsation receiving systems installed by R. L. Dowden from Otago University, New Zealand, in a field station of UiT in Lavangsdalen, about 17 km from the heater site, together with theory
and modelling, the authors could explain the characteristics and mechanisms involved in the generation process of these low frequency waves. The first author became involved in these experiments as a post-doctoral scientist in 1981 after completing a doctoral degree on VLF wave research at Otago University. ELF/VLF wave propagation experiments were also performed with satellite receivers in the ionosphere and magnetosphere and attempts (unsuccessful) at conjugate wave reception in the Australian Antarctic base in Mawson. Later collaboration with R. Barr from New Zealand explored Earth-ionosphere
waveguide propagation and evaluation of different modulation techniques to explore the efficiency of such wave generation, both experimentally and with modelling.

Lavangsdalen was also the site of the accidental discovery of another important phenomenon, stimulated electromagnetic emissions (SEE), consisting in the generation of secondary HF waves due to plasma processes in the ionosphere under the action of the powerful pump wave (Thidé et al., 1982; Stubbe et al., 1984). This discovery opened up a major research area
at all heating facilities (e.g. Leyser, 2001).

Early experiments were also performed with the local partial reflection experiment (PRE) from UiT, as well as HF diagnostics from the group at University of Leicester led by Tudor Jones. Research from this group led to many theses and papers by Terry Robinson, Alan Stocker and Farideh Honary for example. Naturally the EISCAT UHF radar and later the VHF radar when it came on line in 1985, were used as a new diagnostic instruments. Much scientific interest was in ISR
experiments with heating where the Earth's magnetic field was near vertical in contrast to the interesting results coming out of the Arecibo facility where the magnetic field was near 45°. For near field-aligned HF pumping the Langmuir turbulence results were expected to be much stronger and indeed the experimental results exceeded expectations and provided a source of controversy. There was fruitful exchange between theory and experiment from these experiments. For these ISR diagnostics, which usually involved high time and spatial resolution observations of strong coherent signals induced in the
ion and plasma line spectrum, special modulations and detection algorithms had to be developed with very different requirements from the usual incoherent scatter measurements of the undisturbed plasma. These programs were developed

**History** of Geo- and Space **Sciences** Discussions

mostly by Harry Kohl and Terrence Ho from MPAe. Collaborations were developed with researchers from many countries. These include Tor Hagfors, Brett Isham, Frank Djuth, Mike Sulzer, Shanti Basu. There were several cooperative projects with scientists from the former Soviet Union in various institutes such as the Polar Geophysical Institute in Murmansk,

IZMIRAN near Moscow, and the Institute of Radio Astronomy in Ukraine. Some campaigns after 1990 involved scientists bringing with them diagnostic instruments from the former Soviet Union, which was impossible in the days of the cold war.

From the start of project Heating it was planned to fly rocket instrumentation through the heated region (Stubbe et al., 1978) from the Norwegian rocket base at Andøya. The principle investigator for this HERO (HEating ROckets) project was Dr.

Gerhard Rose from MPAe, with collaborators from the Fraunhofer Institut für Physikalisch Messtechnik (Institute for Physical Measurement Techniques) Freiburg, Germany, the Norwegian Defense Research Establishment (NDRE) and the University of Oslo. The HERO project was the first project especially designed to measure in situ the HF-generated Langmuir waves and their influence on the surrounding plasma. One of the four flights flown in 1982 resulted in important measurements of the HF wave field strength, electron and ion temperatures, and supra thermal electrons excited by the heater

(Rose et al., 1985), but unfortunately the EISCAT radar was not operational during this flight. A special modification to one of the antenna arrays had to be made in order that a rocket could fly through the heated region, since rockets could not be flown over the Norwegian mainland. Phase delay coaxial feed lines were constructed for the mid-frequency Array-2 such that the heater beam could be tilted 13° westwards towards the planned rocket apogee, in addition to a tilt of 7.5° northwards achieved by normal phasing of the transmitters. These lines could be switched in and out within a few minutes, in the same

way that the transmitters could be switched between different antenna arrays. This westward tilting ability was also used after the rocket campaigns for some satellite radio beacon experiments, but the coaxial lines of this phase shifter were later removed to reduce maintenance. The remaining switches and control hardware may still have a useful future, however, as will be mentioned below.

The most important results from the first phase of operation of the heating facility, led by researchers from MPAe, were

summarised in Stubbe (1996).

## 4. A storm and antenna array reconfiguration

The three antenna arrays were all used depending on which frequency was optimum for the particular science goal. Many experiments where plasma instabilities are excited required pump frequencies near the O-mode critical frequency in the F

layer which has a daytime maximum that varies with solar radiation and the solar cycle. Other experiments, which rely on maximum Ohmic heating of the lower ionosphere are best with lower frequencies. So Array-1, the lowest frequency and largest array, was used mostly for VLF/ULF modulation and other D region experiments. On 25 October 1985 extremely high winds during a storm seriously damaged about 75% of the 36 antenna towers in Array-1, leaving all the wooden masts

intact. None of the towers in the other two arrays were damaged. The aluminium towers in all three arrays were of identical construction, differing only on their height, being 12 m in Array-3, 16 m in Array-2 and 23 m in Array-1. They were anchored only at the top and bottom with no guy wires in between, which was probably the reason for the failure: the Array-1 towers were too tall and hence not rigid enough such that they bent under the force of the wind.

Rather than rebuild the array in its original form, it was decided that the lower frequency band could be dispensed with since ULF-VLF wave generation experiments could also be done at higher frequencies and at these lower frequencies there would
be too high Landau damping of the Langmuir waves generated by parametric instabilities to make them interesting. On the other hand, one lost the opportunity to examine effects at the second gyro-harmonic and one lost flexibility in frequency choice near around solar minimum when the critical frequencies are often very low. It was decided to add to the existing 23 m high wooden masts, 120 wooden masts of 12 m height and install 144 crossed full-wave dipole antennas in a 12 x 12 configuration for the 5.5-8 frequency band, within the same area as the 6 x 6 original low frequency antennas (see Fig. 2).
This gave the array a gain of 30 dBi which corresponded to an ERP of 1200 MW compared to the 24 dBi of Array-3 such that Array-1 was sometimes called the super heater. The resulting beam was therefore narrower than that of the other two arrays and it could not be tilted as far off zenith. Experiments with the rebuilt array started in 1990 and continued under MPAe leadership until the transfer to EISCAT in January 1993. Although no new phenomena were discovered with the higher gain array, the higher power density was useful for many experiments, especially in the D region or mesosphere as
later experiments were to show. A more detailed description of the HF facility, as it was then, is given in Rietveld et al. (1993).

## 5. Transfer to EISCAT and user expansion

After nearly 10 years of successful discoveries and exploration, it was decided that MPAe would end the Heating project. This was suggested by Axford since the emphasis of research at the institute was increasingly moving towards space-based
instrumentation, and ionospheric research activity was decreasing. This was in line with a general policy within the Max-Planck Society to fund research projects only for a limited time and to start new ones. A transfer of the facility to the EISCAT Scientific Association was offered, and after a scientific case (Robinson et al., 1989) was prepared by a group of interested scientists led by Prof. Terry Robinson, the facility was formally transferred to EISCAT in January 1993. The first author, who was employed at EISCAT since 1987, was responsible for running the facility from 1993 until December 2020,
and initially two engineers were dedicated with the operation and maintenance of the HEATING division of EISCAT. Gradually, the staff at Ramfjordmoen shared the tasks necessary to operate the ISRs and HEATING as required by changing user demands of the different facilities. For example, much effort was spent in building the EISCAT Svalbard Radar (ESR) in the early to mid 1990s which diverted science interest and operations to the polar cap region.

The transfer to EISCAT resulted in a larger user group continuing heating experiments which used nominally 200 hours of
heater time per year, but which varied between 100 and 300 hours depending partly on the solar cycle. Fewer experiments
were possible when the F region critical frequency was low during solar minimum. Most experiments were in conjunction
with the ISR as the major diagnostic instrument in many cases, or as one of several diagnostics. Improvements in diagnostic
instrumentation and the deployment of new instruments, as well as improved radar coding, modulation and data storage
associated with advances in computing technology led to new discoveries that were impossible or difficult to achieve in the
first decade of operation. Some examples are the discovery that heating the lower ionosphere can weaken or supress polar
mesospheric summer echoes (PMSE) observed by the VHF (224 MHz) radar (Chilson et al., 2000), and the production of
light emission from the heated ionosphere (Brändström et al., 1999). These two very different areas of research, namely
study of the mesospheric dusty plasma and energetic electron acceleration, have remained as major topics of research up to
the present time.

A technique developed in the former Soviet Union which uses only the powerful HF waves to measure ionospheric and
atmospheric parameters by the production of artificial periodic irregularities (API) (Belikovich et al., 2002), was successfully
applied in the auroral ionosphere for the first time by combining the heating facility as a transmitter and the Dynasonde as a
receiver (Rietveld et al., 1996b). Later, more sophisticated experiments by Vierinen et al. (2013) showed how this technique
is particularly interesting and particularly promising for studies of the mesosphere.

The heating facility remained essentially unchanged through the 1990s. An operating licence was obtained to allow
frequency stepping experiments around harmonics of the gyro-frequency. There was an upgrade of the computer from the
original Commodore PET to a Microsoft Windows based personal computer system in 1999 where the original BASIC
control program was converted to Hewlett Packard BASIC, and the more modern computer allowed RF synthesizer and
transmitted HF parameters to be stored in a digital log. Previously the transmitter settings were recorded largely in a hand-
written log book. There were minor changes made to some of the control system, such as a programmable step change in the
control grid bias voltage during long (> ca. 1 s) RF off intervals such that the quiescent current in the transmitter tubes
dropped from about 6 to 1 A (at 10 kV in each of 12 transmitters!) to save on electricity power consumption. One saves most
on power consumption when the high voltage is switched off so that no quiescent current flows through the tube, but this had
to be done manually by pushing 12 buttons that actuate 12 large relays, something that is undesirable for non-transmitting
intervals of a few seconds or tens of seconds or a few minutes, which are rather commonly used modulation periods. The
cost of electric power for heating operation, especially when experiments required Heating together with the VHF and UHF
radars, was a major economic concern in the 1990s at a time when there were some EISCAT associates who were
advocating the closure of the HF facility to save money.

Around 2005 plans were made to upgrade the synthesizers to direct digital synthesis (DDS) and associated computer control
to a unix-based system. Apart from replacing ageing hardware, a major motivation was to allow fast frequency changes of



the HF pump wave which were increasingly requested in order to examine the ionospheric response to HF pumping at and
near harmonics of the ionospheric gyro-frequency. Previously, frequency changes required a several-minute long tuning and
phasing procedure under computer control, because the HP synthesizers started with a random phase value for any frequency
change. The digital system would allow setting of phases to any desired values practically instantaneously. The final system,
which was taken into regular use in 2009, used some hardware and much software that the EISCAT ISRs had implemented
in the mid-1990s when the ESR was built. This upgrade was a major effort involving expertise of EISCAT staff from
Tromsø as well the other two mainland sites, EISCAT headquarters in Kiruna, Sweden, and Sodankylä in Finland. This
upgrade and further improvements to the Heating system are described in Rietveld et al., (2016). From 2012 even more
functionality has been developed such that the status of many transmitter and array parameters that were only indicated by
lights or controllable by buttons are now monitored and set by computer. Figure 6 shows the console in the control room
before the upgrade. The main difference to that after the upgrade is the replacement of the 12 original synthesizers in the
central part by large computer screens which are now used to control and monitor the facilities operation and observe some
scientific results in real-time.

In 2013 a modification was made to the coaxial switches that fed Array-3 to allow receivers to be connected to that array.
The motivation for this was to try and receive magnetospheric echoes, for example from ion-acoustic turbulence excited by
auroral processes such as has been observed by VHF and UHF incoherent scatter radars (Rietveld et al., 1996a). Previously,
related experiments had been tried using Heating as a transmitter and the large HF radio telescope, UTR2, in Ukraine as a
receiver but without results. Since the modified Array-1 and Array-3 cover the same frequency range, one could transmit on
Array-1 and receive on Array-3, albeit with different antenna gains and hence beam widths. The first version of a receiver
connected to Array-3 for radar work is described in Rietveld et al. 2016, where fixed-length phasing cables were used to
combine the signals from the six rows of orthogonal antennas into two receiver channels. Since 2017 the individual rows of
antennas are each connected to a digital receiver allowing beam forming of the received signal in the north-south plane. This
receiving system seems to works well for mesospheric echoes but echoes from the magnetosphere have never been detected
to date. Using Array-3 as a receiving antenna has some weaknesses such as the aluminium connectors in the feeder lines
where an oxide layer may adversely affect weak radio signals, but which is burnt through by the powerful radio wave on
transmission for which it was designed.

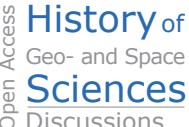

Figure 6: The heating facility console in the control room as it was in 2007, with the first author at the controls. Six columns of meters, lights and push buttons on each side show the status and are used to control each of the 12 transmitters. The twelve commercial RF synthesizers in the middle of the console have been replaced by digital synthesizers in the transmitter

hall and the space is now filled by large computer screens. (Photo M. T. Rietveld).

The Heating facility was only intended to operate for a limited time of about 10 years, so that 40 years after construction it was inevitable that some parts of the system aged to a critical point or that spare parts become unobtainable. One key component is the transmitting tube in each of the twelve power amplifiers. The original tube was no longer produced after

1980 but a good number of spares allowed operation at near full power level until recently. Very few tubes failed completely, but after about 12000 hours of filament-on time over the lifetime of Heating, several were slowly delivering less power. A few tubes were sent to firms in USA for rebuilding but the success rate was poor. In searching for an alternative tube that required minimal modification to the transmitters it was found that the RS2054SK tetrode was almost compatible

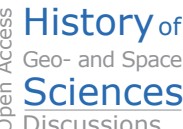

with the existing transmitter and this type was still manufactured in Europe and in China. Although it was a drop-in
replacement in the tube socket, the new tube required a different filament voltage and a slow ramping up and down of the
voltage so that several modifications to the transmitter had to be made. In 2018 the first of the new tubes entered operation
and at present two transmitters use the new tubes with a third ready to be similarly modified.

Another ageing problem that first appeared around 2008 was in the modified Array-1. The feed cables to the antennas in this
array were different from the original design in that instead of towers made of aluminium coaxial cable feeding each
antenna, commercial twin wire flexible cable was used. Starting in 2008 an increasing number of the 288 feed points at the
antenna failed during transmission through burning insulation and fire, often resulting in the collapse of the whole feed point
and antenna to the ground and sometimes resulting in a dramatic grass fire. The cause of this failure was a mystery for a long
time, and a total of 17 feed cables and centre part of the antennas had to be laboriously repaired by 2017. The cause was
found in 2015 to be metal fatigue in flexible twin-wire braided copper cable where it was anchored to the fixed centre of the
antenna, with the rest of feed cable from near the ground free to move slightly in the wind for about 20 years. Many of the
anchoring guy ropes which should have minimized movement of the cable in the wind had broken and were not repaired.
The wind-induced movement of the cable caused many of the braids to break such that the cross sectional area of the cable
was reduced to the point that the high power RF caused overheating or arcing across the final break resulting in the burning
of the insulation. The solution was to bypass the anchor point with a short piece of wire crimped to the feed wires on each of
the 144 antenna centres on the 12 m tall masts, a job which took several summer seasons and required EISCAT to buy a lift
to safely implement the repairs. It is hoped that this solution is robust enough for the remaining lifetime of the heating
facility. Figure 7 shows the repair work in Array-1, where the different feed cables compared to those in Fig. 3 are clearly
visible.



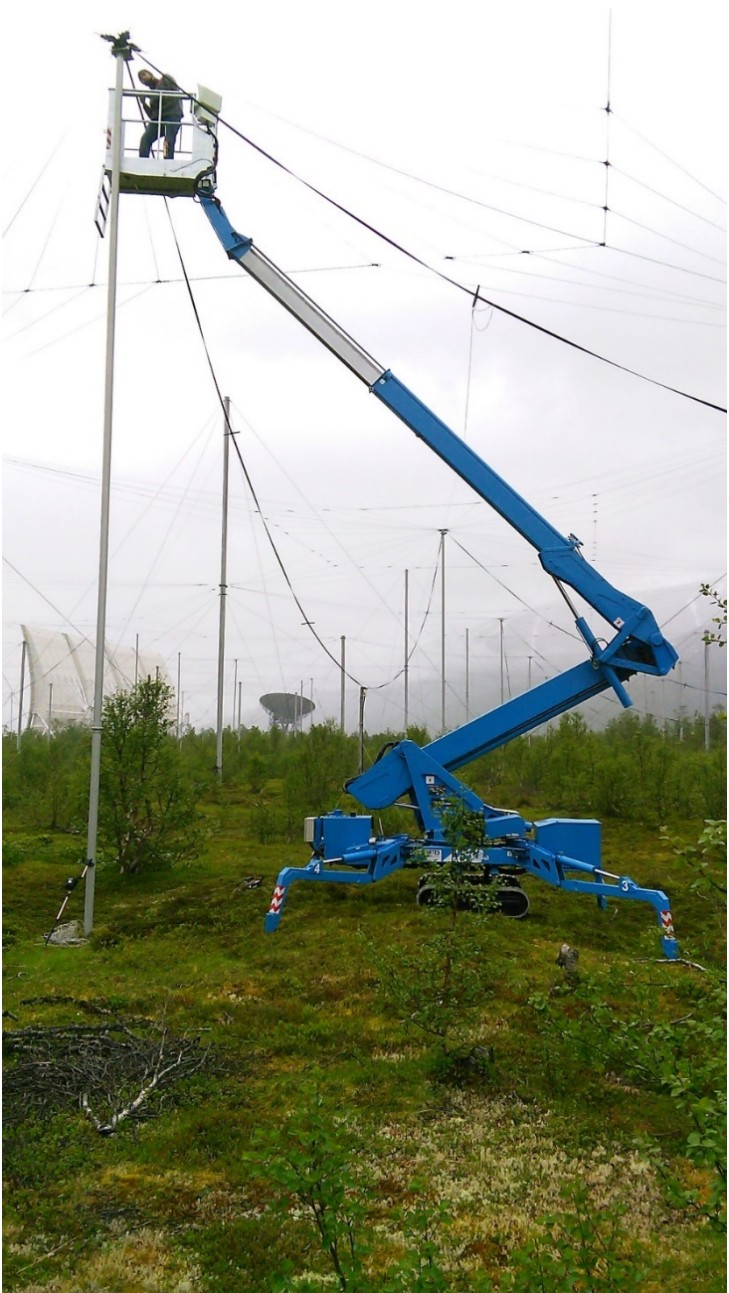


Figure 7: Repair work to one of the 12 m high antennas out of 144 such antennas in Array-1 to bypass existing and potential weak points in the twin-feed cable connection to the antenna centre. The original 23 m wooden masts are visisble in the background. Note the different support mast and feed-lines compared to the original design as used in Arrays-2 and -3 shown in Fig. 3. (Photo M. T. Rietveld)


## 6. Present status and future

The main hardware of Heating, the transmitters, feed lines and antennas, have remained essentially unchanged since 1990, apart from the computer control and RF synthesizers upgrades described above. The user community has changed with time, with some users and groups changing field but there are also new users entering the field. With the closure of other facilities like HIPAS in Alaska (Wong et al., 1990) and SPEAR on Svalbard (Robinson et al., 2006) and the catastrophic failure of the towers supporting the Arecibo ISR antenna feed system and which were also used to support the heating antennas, there is only HAARP in Alaska (Pedersen and Carlson, 2001) and SURA in Russia (Belikovich et al., 2007) which have working HF ionospheric heating facilities. Neither of these installations have incoherent scatter radars as diagnostic instruments, however, which makes the Tromsø HF heater a unique and valuable facility for the world.

Groups from all the EISCAT Associates have been regular users of the heating facility. In recent years, researchers from China, where the China Research Institute of Radiowave Propagation (CRIRP) became an EISCAT Associate in 2007, have become important regular users. A large international community of users have been able to use the Tromsø HF facility, especially in the last decade when non-EISCAT Associates could either buy time or apply for a limited number of free hours on either or ISR or heater or both, through a peer-review system. An excellent example of fruitful scientific results heating experiments by a non-EISCAT associate is a 25-year collaboration with a group from the Russian Arctic and Antarctic Research Institute (Blagoveshchenskaya et al., 2020). Other long-term users were researchers from the Polar Geophysical Institute, in Murmansk, Russia, and from the Institute of Radio Astronomy in Ukraine. A description of the various scientific results that have been obtained from the heating facility is beyond the scope of this paper. The number of accumulated publications from the Tromsø heating facility amounts to more than 480 which are listed on the EISCAT publications web page (https://eiscat.se/scientist/publications/heating-publications/). Streltsov et al. (2017) discuss many of the physical problems that are topics of present and future research in the field of active experiments using high power radio waves. Some of the interesting phenomena to explore are narrow band SEE (stimulated Brillouin scatter), artificial ionization, unexplained X-mode effects, and the irregularities postulated to explain wide altitude ion line enhancements sometimes known by the acronym WAILES (Rietveld and Senior, 2020).

The Tromsø heating facility has not been overly troubled by adverse publicity or conspiracy theories like the HAARP facility in Alaska has. One reason that HAARP had this problem is possibly because it was built and operated for many years by military organisations and some of the experiments conducted there were classified. The experiments conducted at the Tromsø facility were always open and all publications resulting from it appear in the open literature. Nevertheless, over the years there have also been exaggerated and false claims made about some types of experiments that are possible with heating facilities like that in Tromsø.

The site in Ramfjordmoen is about to undergo major change when the EISCAT UHF and VHF radars are decommissioned and EISCAT-3D, the next generation incoherent scatter radar (McCrea et al., 2015), starts operation with the core site in nearby Skibotn. Since the retirement of the first author, the heating facility is being led and run by Erik Varberg. The heating facility is planned to remain in operation for experiments with the new radar which will offer unprecedented insights into the HF-induced phenomena. The improved spatial resolution and the ability to quickly steer the beam of the new radar electronically or to have multiple beams should help probe the horizontal spatial properties of HF-induced irregularities. There is one disadvantage in not having the HF-facility and the radar co-located, namely the radar cannot probe field-aligned along the heater beam in the F region. This problem may be overcome by resurrecting, in slightly modified form, the east-west tilting hardware built for the HERO rocket campaign in the 1980s mentioned earlier. Most of the switching hardware still exists but new aluminium coaxial phasing cables would need to be made and installed. The possibility of building a new heater nearer Skibotn is also being investigated.

**Author Contribution**

M. T. R. Wrote most of the manuscript and P. S. provided additions and corrections.

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
