# Peer review of "History of the Tromsø ionosphere heating facility"

_History of Geo- and Space Sciences, 2021_

## Referee Comment (RC1)

**Review of History of the Tromsø Ionosphere Heating facility**

Michael T. Rietveld and Peter Stubbe

The first eighteen pages of this manuscript are clear, concise, and tell a very interesting story of the evolution of the Tromsφ ionospheric heating facility. It provides excellent detail on the design of the facility and the specific nature of the parts used. Particularly interesting are the description of the problems encountered and the solutions developed particularly in light of the newness of the design and the availability of parts in the late 1970's. Also the initial use of HP3325A synthesizers, the Commodore PET and the Texas Instruments TM 990 to generate the RF waveform was innovative, but with increasing time the electronics evolved to direct digital synthesis and a UNIX computer system.

The loss of the low frequency Tromsφ array on 25 October 1985 must have been disheartening, but it led to the decision to develop and build a higher frequency (5.5 – 8.0 MHz), 30 dBi gain, 1.2 GW ERP array. This array led to many observations and discoveries that would not have been possible with the original three arrays. Finally, the very long 40-year lifetime of the Tromsφ heating facility was achieved in part by replacing the old transmitter tubes with new tetrodes (RS2054SK). Hopefully, the system will remain in operation for at least 20 more years.

**Page 19, Lines 383-390 Arecibo discussion**

"The main hardware of Heating, the transmitters, feed lines and antennas, have remained essentially unchanged since 1990,apart from the computer control and RF synthesizers upgrades described above. The user community has changed with time, with some users and groups changing field but there are also new users entering the field. With the closure of other facilities like HIPAS in Alaska (Wong et al., 1990) and SPEAR on Svalbard (Robinson et al., 2006) and **the catastrophic failure of the towers supporting the Arecibo ISR antenna feed system and which were also used to support the heating antennas**, there is only HAARP in Alaska (Pedersen and Carlson, 2001) and SURA in Russia (Belikovich et al., 2007) which have working HF ionospheric heating facilities. Neither of these installations have incoherent scatter radars as diagnostic instruments, 390 however, which makes the Tromsø HF heater a unique and valuable facility for the world."

The above paragraph groups Arecibo Observatory with two other facilities that will never return to operation. It has only been a little more than a year since the collapse of the Arecibo radar, and the emptying of the 305 m dish of 900 tons of metal and debris takes time. In addition, the writing of a proposal(s) for a new state-of-the-art ISR, planetary radar, and radio astronomy system at Arecibo is a research project unto itself. Also there is an agonizingly slow approval rate by the U.S. National Science Foundation on large-scale competitive projects such as this. Below is a more precise description of the status of Arecibo than discussed in this paper.

On December 1, 2020, after 57 years of usage, Arecibo Observatory's 900 ton platform containing transmit/receive feeds fell ~150 m and crashed into the 305 m diameter reflector dish. This halted ISR, HF heating, planetary radar, and radio astronomy observations at the Observatory. Full or partial recovery plans are currently under consideration by the U. S. National Science Foundation. Complete decommissioning appears unlikely. A modest HF facility is currently being constructed at Arecibo to keep HF heating research moving forward.

**Page 19, lines 408 – 414 HAARP**

"The Tromsø heating facility has not been overly troubled by adverse publicity or conspiracy theories like the HAARPfacility in Alaska has. One reason that HAARP had this problem is possibly because it was built and

operated for many years by military organisations and some of the experiments conducted there were classified. The experiments conducted at the Tromsø facility were always open and all publications resulting from it appear in the open literature. Nevertheless, over the years there have also been exaggerated and false claims made about some types of experiments that are possible with heating
facilities like that in Tromsø."

In the U. S. everyone is entitled to their own opinion.  There has always been a controversy as to whether the Federal Senators from Alaska should seek projects that will bring jobs and commerce to Alaska as HAARP did or whether they should do this on their own.  Most U.S. senators seek out Federal funds for the populations in their States.  The HAARP area between Gakona and Glen Allen is very peaceful.  There are no protests, and every year there is an 'open house' where everyone can come to the site and see first hand what all of the features of the facility are, ask questions, and most importantly get their picture taken in front of the huge HF antenna.  There are newspapers, TV shows (e.g. HAARP 'reprograms people'), and books (e.g. Angels Don't Play This HAARP) that are HAARP negative, but no one really believes any of this.  If they did they would request observing time to reprogram their employees.

People from the relevant "military organizations" discussed above include some of our colleagues such as Paul Bernhardt, Todd Pedersen, and many others.  Everyone publishes all of the data of interest from HAARP, and there are many HAARP publications.  For a complete listing contact Paul Bernhardt.  There is only one classified project that this reviewer is aware of.  It was orchestrated by a University professor who brought active high-level military people to HAARP.  This project received observing time similar to that of other investigators, and no one had to leave the HAARP facility when the "classified" experiment was conducted.  In fact most of the scientists present understood the objectives and how one would accomplish the key goal.  The concept that there were many classified experiments conducted at HAARP is not true.  Below is a description of HAARP that is suitable for publication.

HAARP was originally constructed and operated with funds secured by senior research science managers of the U. S. Air Force and Navy.  Outside investigators proposed experiments and received observing time cost free. Many journal publications were produced because no facility usage fees were involved.  Subsequently all HAARP operations were turned over to the University of Alaska, Fairbanks in August 2015. However, because of the high cost of operations particularly for a university, outside investigators are required to contribute to their observing costs on a pay-for-use basis.  Thus funds from grants and contracts or from internal sources are necessary to support the research.

---

## Author Response (AR1)

We thank the referees for their comments.

The line numbers refer to the revised text, tracked changes, submitted here.

**In response to referee 1** we have incorporated the reviewer's suggested description of the Arecibo heater status in the first paragraph of the revised section 7 (lines 390-399).

In the section on conspiracy theories where HAARP is mentioned, we have re-written that section in a more neutral way, but we did not include the referee's suggested description of HAARP since we find the funding details of other facilities not relevant for this paper (lines 417-423).

**In response to referee 2**, we have incorporated all suggestions made.

Line 36: We believe this form of reference to a whole journal issue is correct.

Dipl. Phys. is now explained in parentheses.

The use of "guy" is indeed correct. Among other meanings it includes: "A rope, cord, or cable used to steady, guide, or secure something. tr.v. guyed, guy·ing, **guys** To steady, guide, or secure with a rope, cord, or cable. [Partly from Middle English gie, guide, **guy** (from Old French guie, from guier, to guide; see weid- in Indo-European roots)   (www.thefreedictionary.com)

Line 259: The units of the frequency range were omitted. They have now been inserted.

**Detailed changes**:

Oxford spelling has been chosen and made consistent.

Section numbering has been changed so that the introduction is section 1 and subsequent sections numbers are incremented by 1.

The abbreviation "IS" replaces "incoherent scatter" in most places. In addition, to make the text more consistent the abbreviation ISR has been replaced with "IS radar".

Lines 53-54: An explanation of the degree "Dipl. Phys." is given in brackets.

Lines 71-72 and references (line 521-522): The reference to Wannberg (2021) is updated to the published version in 2022.

Fig. 2 caption (line 117) "(small red crosses)" is added to the description.

Line 195: The relevant height region is given.

Line 221: One more person, Cesar La Hoz, is added to the list of important collaborators.

Line 261: "around" which is redundant is removed.

Line 263: "MHz" has been added.

Line 403: One instance of "users" has been changed to "scientists"

Line 405: change to "peer-review programme" instead of "system"